# Genetic Mapping to Detect Stringent QTLs Using 1k-RiCA SNP Genotyping Platform from the New Landrace Associated with Salt Tolerance at the Seedling Stage in Rice

**DOI:** 10.3390/plants11111409

**Published:** 2022-05-26

**Authors:** Sheikh Maniruzzaman, Mohammad Akhlasur Rahman, Mehfuz Hasan, Mohammad Golam Rasul, Abul Hossain Molla, Hasina Khatun, Salma Akter

**Affiliations:** 1Plant Breeding Division, Bangladesh Rice Research Institute (BRRI), Joydebpur, Gazipur 1701, Bangladesh; skmonir85@yahoo.com (S.M.); hasinabrri09@gmail.com (H.K.); 2Department of Genetics and Plant Breeding, Bangabandhu Sheikh Mujibur Rahman Agricultural University, Salna, Gazipur 1706, Bangladesh; mehfuz@bsmrau.edu.bd (M.H.); mdgolam60@yahoo.com (M.G.R.); 3Department of Environmental Science, Bangabandhu Sheikh Mujibur Rahman Agricultural University, Salna, Gazipur 1706, Bangladesh; ahmolla60@gmail.com; 4Plant Physiology Division, Bangladesh Rice Research Institute (BRRI), Joydebpur, Gazipur 1701, Bangladesh; salmabrri@gmail.com

**Keywords:** new QTL, SNP genotyping, salt tolerance, *indica* germplasm, *Oryza sativa* L.

## Abstract

Rice is the world’s most important food crop, providing the daily calorie intake for more than half of the world’s population. Rice breeding has always been preoccupied with maximizing yield potential. However, numerous abiotic factors, such as salt, cold, drought, and heat, significantly reduce rice productivity. Salinity, one of the major abiotic stresses, reduces rice yield worldwide. This study was conducted to determine new quantitative trait loci (QTLs) that regulate salt tolerance in rice seedlings. One F_2:3_ mapping population was derived from a cross between BRRI dhan49 (a popular but sensitive rainfed rice variety) and Akundi (a salt-tolerant rice landrace in Bangladesh used as a donor parent). The 1k-Rice Custom Amplicon (1k-RiCA) single-nucleotide polymorphism (SNP) markers were used to genotype this mapping population. After removing segregation distortion and monomorphic markers, 884 SNPs generated a 1526.8 cM-long genetic linkage map with a mean marker density of 1.7 cM for the 12 linkage groups. By exploiting QGene and ICIM-ADD, a sum of 15 QTLs for nine traits was identified in salt stress on seven chromosomes. Four important genomic loci were identified (*qSES1*, *qSL1*, *qSUR1* and *qRL1*) on chromosome 1. Out of these 15 QTLs, 14 QTLs are unique, as no other study has mapped in the same chromosomal location. We also detected 15 putative candidate genes and their functions. The ICIM-EPI approach identified 43 significant pairwise epistasis interactions between regions associated with and unassociated with QTLs. Apart from more well-known donors, Akundi serves as an important new donor source for global salt tolerance breeding initiatives, including Bangladesh. The introgression of the novel QTLs identified in this study will accelerate the development of new salt-tolerant varieties that are highly resistant to salt stress using marker-enabled breeding.

## 1. Introduction

Rice (*Oryza sativa* L.) is the world’s most important staple food crop, consumed by more than half of the world’s population [1,2,3]. Rice is a tropical diploid (2n = 2x = 24) glycophyte that serves as the model crop for cereals [4]. Rice breeding has always been used for maximizing yield potential. However, numerous abiotic stress factors, such as salt, cold, drought, high night temperature [5] and heat, significantly reduce rice productivity.

Salinity is one of the key abiotic stresses that poses a severe global challenge to food security [6]. Because rice is a salt-sensitive plant, increasing salinity levels affects its productivity. The extent of salinity in salinity-affected areas is continuously increasing in under the conditions of climate change [7,8]. Globally, salinity affects around one-third of land area [9]. Rice seedlings are particularly susceptible to biotic and abiotic stresses. High salinity inhibits plants’ uptake of water and nutrients from the soil, resulting in seedling growth, development and establishment in the main field, which affects and ultimately reduces production. The increase in rice productivity in saline-prone areas is inevitable [10]. Salinity stress is a serious constraint on rice farming, due to the vulnerability of rice throughout its young seedling stage through to reproductive development [11,12,13,14]. As a result, breeding programs in coastal zones must prioritize the development of salt-tolerant rice genotypes.

Several studies revealed that the genetics of complex traits such as salt tolerance are influenced by polygenes and QTLs [8]. Different QTL regions controlling various salinity-related traits in rice were identified through genetic mapping using segregating or permanent mapping populations. Three significant QTLs were identified on chromosomes 1, 8, and 10 using the F2 mapping population [15]. Eleven E-QTLs and eleven M-QTLs were reported by Wang et al. [16] for the salt tolerance indices from the RIL mapping population. Mohammadi et al. [17] reported that 35 QTLs were found for 11 characteristics in an F_2_ mapping population. Hossain et al. [12] reported that, by utilizing 141 SSR markers in an F_2:3_ mapping population, a maximum of six QTLs with LOD values in the range from 3.2 to 22.3 were revealed on chromosomes 1, 7, 8, and 10. QTLs responsible for Na^+^ concentration, K^+^ concentration, Na^+^/K^+^ ratio, and survival were detected on the long arm of chromosomes 1 and 3 [8]. Moreover, six pairwise epistatic interactions between QTL-linked and QTL-unlinked areas were observed. A total of 151 trait-marker associations were identified, which were located on 10 different chromosomes of rice arranged in 29 genomic regions under salt-stress conditions [18].

Exploring effective traits for identifying useful alleles in rice germplasms is important. Akundi, a salt-tolerant Bangladeshi *indica* landrace, accumulated lower Na^+^ and comparatively higher K^+^ in leaves, resulting in a reduced Na^+^/K^+^ proportion. This landrace significantly restricted sodium flow to the shoots from the roots, which was recognized as a potential source of salinity tolerance [14]. Apart from the more well-known donors—such as Capsule [8], Nona Bokra, and Pokkali [19]—Akundi [14] may serve as an alternative of salt tolerance for global breeding programs, including those in Bangladesh. Pokkali’s better adaptability for salinity is related to two distinct characteristics: its ability to maintain a minimal Na^+^/K^+^ balance in the shoot tissue, and thus its healthy growth rate in saline environments, which aids in dispersing the salt and lowering cytotoxic stress within the tissue [19,20,21]. The mapping of quantitative trait loci is critical for advancing our understanding of the distribution and genetic structure of component characters [22]. Moreover, it enables the genetics of complex traits such as salinity to be explored and facilitates the marker development of the traits of interest, stacking useful alleles into promising breeding lines using designed QTL pyramiding. The main objective of the study was (i) to identify new quantitative trait loci (QTLs), which regulate salt tolerance in the seedling stage of rice, and (ii) to detect epistasis/digenic interactions through identifying E-QTLs.

## 2. Results

### 2.1. Investigating the Salt Stress Responses 

A total of 91 F_2:3_ individuals were derived from a cross of BRRI dhan49/Akundi and evaluated in a phytotron under salt-stress conditions. The plants were salinized 14 days after seeding by adding NaCl to the culture solution until an electrical conductivity of 12 dS m^−1^ was attained. The F_2:3_ population had sufficient genetic variability from each other, which is discussed below. 

#### 2.1.1. Characterizing Agronomic Traits under Salt Stress

Two parents demonstrated significant reactions to salt injury, with BRRI dhan49 (BR49) being extremely sensitive (SES score = 7) and Akundi being tolerant (SES score = 4). Figure 1a depicts the frequency distribution of the mapping population and its parents. The average survival for Akundi was 95% (range: 90–100%), while the average survival rate for BR49 was 50% (range: 40–60%). This trait exhibited a negative skew (skewness = −1.53). Around 45% of individuals in the population had a higher survival rate (96–100%) than the donor parent, while only 5% had a higher sensitivity than BR49 (Figure 1b). The average shoot length of Akundi was 55.00 cm (range: 53.60–56.40 cm), whereas BR49 had an average shoot length of 28.85 cm (range: 24.70–25.00 cm). Shoot length variability was also significant among F_2:3_ progenies (range: 28.50–52.70 cm). Additionally, the frequency distribution exhibited a heterogeneous dispersion with a skewness of 0.45. (Figure 1c). The dry weight of the shoots ranged between 0.03 and 0.22 g: 0.22 g and 0.05 mg for Akundi and BR49, respectively (Figure 1d). Out of the 91 F_2:3_ progenies, 52 progenies had shoot biomass in the range of 0.10–0.20 g, while only 10 progenies had a shoot biomass of less than 0.07 g, comparable to that of BR49. This distribution was positively skewed (0.23). The mean root length of Akundi was 13.35 cm (range of 12.90–13.80 cm); while that of BR49 was 11.55 cm (range of 11.00–12.10 cm) (Figure 1e). The root length variation was also large in the F_2:3_ progenies (range: 9.05–20.60 cm). The frequency distribution was also positively skewed (0.18).

#### 2.1.2. Physiological Traits under Salt Stress

The SPAD value of Akundi was 31.43 and ranged from 30.70–32.15; BR49 was 23.65 and ranged from 22.8 to 24.5 (Figure 1f). The highest SPAD value (more than 31.43) was observed in 12 individuals and the lowest SPAD value (less than 20.00) was in 10 individuals. The data were negatively skewed (−0.25).

The lowest Na^+^ concentrations (between 0.02 and 0.09 mmol g^−1^ dwt) were observed in 45 individuals, while the highest concentrations (greater than 1.5 mmol g^−1^ dwt) were observed in 13 individuals (Figure 1g). The remaining individuals had an intermediate accumulation of Na^+^. The Na^+^ concentrations of Akundi (0.07 mmolg^−1^ dwt) and BR49 (0.10 mmolg^−1^ dwt) were also significantly different.

Potassium concentration was negatively skewed (−0.30). Two individuals had a low (less than 0.31 mmog^−1^ dwt) K^+^ concentration, twenty plants had a high (more than 0.77 mmolg^−1^ dwt) K^+^ concentration, and the rest had a moderate K^+^ concentration. The K^+^ concentrations of Akundi (0.77 mmolg^−1^ dwt) and BR49 (0.31 mmolg^−1^ dwt) were significantly different (Figure 1h).

Akundi and BR49 had a Na^+^/K^+^ ratio 0.09 and 0.33, respectively. In this population, about 58 individuals had Na^+^/K^+^ ratios ranging from 0.04 to 1.8, and 26 progenies had ratios ranging from 1.9 to 3.0, with the remainder having increased Na^+^/K^+^ ratios. The frequency distribution is positively skewed, with a skewness coefficient of 1.16 (Figure 1i).

### 2.2. Correlation Analysis between Traits 

Effective selection strategies for complex (polygenic) salinity tolerance traits depend on the information of the association/relationship between these traits. We observed positive correlations among standard evaluation system scores (SES), shoot length, shoot dry weight, and Na^+^ concentration, as well as significant and negative correlations with root length, SPAD value, and K^+^ concentration (Figure 2). Here, SES and Na^+^ concentration had a positive correlation which showed that, if the sodium concentration increased in plant tissue, then the visual scores (SES score) also increased; the two traits are dependent. SES with SPAD value and K^+^ concentration had a negative correlation, indicating that the visual symptoms decreased as the chlorophyll and K^+^ concentration accumulation increased in the plant tissue. There was a positive and significant correlation between survival rate and shoot length and K^+^ concentration, but a negative correlation between survival rate and SES; therefore, a high K^+^ concentration and low SES score were important parameters for the survival of the plant in stress conditions. The K^+^ concentration had a positive and significant correlation with survival rate, shoot length, shoot dry weight, and root length negative and significant correlation with SES. Therefore, K^+^ concentration is an important factor in plants under stress conditions (Figure 2).

### 2.3. Path Analysis to Assess the Contribution of Traits (Independent Variables) to Salt Tolerance (Dependent Variable) through Partitioning

Shoot length, shoot dry weight, and Na^+^/K^+^ ratio all had a significant positive influence on salt tolerance, whereas survival rate, root length, SPAD value, Na^+^ concentration, and K^+^ concentration all had a negative direct effect on overall tolerance (Table 1). It is truly the case that a lower SES score indicates a greater tolerance for salt, and a lower degree of Na^+^/K^+^ ratio suggests a lower SES score, which is consistent with the correlation (0.218). Through K^+^ concentration, the Na^+^/K^+^ ratio had an indirect influence on tolerance. This demonstrated that the direct negative effect of K^+^ on SES (0.29) was reflective of tolerance. Numerous different traits’ results, such as survival rate, shoot length, shoot dry weight, root length, SPAD value, and Na^+^ concentration were used to calculate indirect path coefficients, which indicated the extent to which these traits indirectly influenced SES scores through their effects on other traits (Table 1). The overall value of indirect effects was determined by adding the indirect path coefficients for each attribute.

The residual effect (R) is how much of a suitable factor for the dependent variable is explained by the residual effect (R), which is calculated as 0.44, as shown in Table 1. This indicated that the variables survival rate, shoot length, shoot dry weight, root length, SPAD value, Na^+^ concentration, K^+^ concentration, and Na^+^/K^+^ ratio together account for 56% of the variations in salinity tolerance.

### 2.4. Marker Segregation and Genetic Linkage Mapping

1k-RiCA SNP markers were used to check marker polymorphism between parents. A total of 884 markers (88.4%) were found polymorphic, which means they were different from each other. This is because both parents (Akundi and BR49) come from an *indica* origin, which means there should be a lot of genetic differences between them. Next, these polymorphic markers were used to genotype the 91 F_2_ individuals. A molecular map was constructed using the Nipponbare/Kasalath genetic map [23] and the physical rice map (IRGSP: International Rice Genome Sequencing Project). The linkage map was 1526.8 cM in length (Kosambi mapping function: [24], with a mean interval length of 1.7 cM (Figure 3). At a LOD of 3.0, a total of four QTLs were identified using single-marker regression (SMR) analysis and 12 QTLs were detected utilizing interval mapping and composite interval mapping (Table 2). QTLs with a strong effect are depicted in Figure 3 on the genetic linkage map. Figure 4 illustrates reflective QTL likelihood graphs (logarithm of odds: LOD curves) for newly identified loci encoding traits associated with seedling-stage salt tolerance.

### 2.5. Salinity Tolerance QTL for Component Agronomic Traits

Two QTLs, *qSES1* and *qSES3,* were distinguished for visual symptoms by the SES score, accounting for 15.6 and 21.6 percent of phenotypic variance, respectively. The QTLs were located at 151.8 on chromosome 1 and 11.3 cM on chromosome 3, respectively, and Akundi contributed the desirable allele. At both loci, the Akundi allele significantly improved phenotypic efficiency and decreased SES visual rating. On chromosomes 1 and 5, three major QTLs for survival were revealed. For the QTLs, *qSUR1,* and *qSUR5.1*, the phenotypic variance was 19.9% and 31.6%, respectively, which was contributed by BR49. However, the QTL *qSUR5.2*, with a phenotypic variance of 14.9%, and the favorable allele was contributed by Akundi. One QTL was recognized on chromosome 1 that was significantly linked with shoot length near the *QSES1-2_2* and the R^2^ value is 30.7% by the IM and CIM method. Two QTLs, *qSDW5* and *qSDW11*, were detected on chromosomes 5 and 11, whose phenotypic variances were 16.3% and 14.0%, respectively. Here, the QTL positions were 82.2 and 41.0 cM and their locus names were chro05_20551103 and chr11_10741559, provided by allele BR49. The QTL *qRL1* was detected from chromosome 1 by the IM method. (SES-LOD value 3.4, survival LOD: 4.3 and shoot length LOD: 7.2, shoot dry weight LOD: 3.5, root length LOD: 3.3) (Table 2).

### 2.6. QTL for Physiological Traits

QTL associated with chlorophyll content was revealed using SPAD readings and it had a substantial LOD score. On chromosome 12, the QTL *qSPAD12*, at a distance of 103.4 cM, explained 16.3% of the phenotypic variation for this trait. One QTL (*qNa6*) on chromosome 6 was found to be strongly correlated with Na^+^ concentration in plant tissue using the IM and CIM methods. A total of 16.8% of the variability in Na^+^ concentration was attributed to the QTL *qNa6.* Two significant QTLs (*qK8*, *qK12*) were detected through IM and CIM for K^+^ concentration. These QTLs are located on chromosomes 8 and 12, accounting for about 21.2%, and 16.3% of the total variation in K^+^ concentration, respectively. Two QTLs, *qNaKR8* and *qNaKR11,* were provided by BR49 and Akundi. (SPAD value LOD: 3.5, Na^+^ concentration LOD: 3.6, K^+^ concentration LOD: 4.7 and 3.5 and Na^+^/K^+^ ratio LOD: 3.0) (Table 2).

### 2.7. Identification of Functional Genes in the QTL Region

The QTLs representing traits SES score (*qSES1* and *qSES3*) were found in the region 38,723,347–38,724,165 and 2,878,828–2,880,890 bp at chromosome 1 and 3 with sixteen and twenty functional genes where the candidate gene was *LOC_Os01g66670* and *LOC_Os03g05770* and the putative function was expressed in proteins, drought-induced proteins, anther and pollen wall remodeling/metabolism proteins contributing to the tolerance of rice to salt stress (Table 3). Three QTLs were found for the trait survival rate (*qSUR1*, *qSUR5.1*, and *qSUR5.2*), which were located at chromosomes 1 and 5, and the candidate genes were *LOC_Os01g22700*, *LOC_Os05g24680*, and *LOC_Os05g05230*, respectively. *LOC_Os01g22700* found in the region of *qSUR1* which is usually considered as an organic cation transporter, affecting root development and carnitine-related responses to stress, as well as numerous biological processes, including transcription, translation, cell signaling, and ion channel activity. The putative gene *LOC_Os05g24680* encoding a retrotransposon protein, putative, Ty3-gypsy subclass, enables plants to cope up with drought stress conditions, and impacts biological processes under stress response was suggested inside a QTL *qSUR5.1* in a 56.0 cM region on chromosome 5. The functional gene *LOC_Os05g05230* corresponding to QTL *qSUR5.2* (10.0 cM) controls the survival rate that is responsible for expressed protein, drought-induced proteins, and anther and pollen wall remodeling/metabolism of proteins that contribute to the tolerance of rice to salt stress. Another functional gene *LOC_Os01g68490* was observed on chromosome 1, encoding tetratricopeptide-like helical and putative genes (abscisic acid responses and osmotic stress tolerance, enabling plants to cope up with adverse environmental conditions) expressed between the chromosomal region 39,794,226–39,799,341 bp inside the QTL *qSL1* at seedling stage. *qSDW5* and *qSDW11* for shoot dry weight were found on chromosomes 5 and 11 in the QTL position 20,966,622–20,969,373 bp and 10,456,526–10,459,838 bp, respectively, where the candidate genes were *LOC_Os05g35310* and *LOC_Os11g18550*, respectively, at the seedling stage. QTL for root length (*qRL1*) found between the region 12,444,630–12,446,150 bp on chromosome 1 with 12 loci, whose putative functions are transposon protein, putative, CACTA, and En/Spm sub-class, significantly contributes to genome size, producing a large number of cDNA sequences in plant tissues and different conditions of stress.

In the chromosomal region 26,377,868–26,379,849 bp of chromosome 12, one functional gene *LOC_Os12g42440* was identified as the chaperone protein dnaJ, responding to NaCl stress and involved in basal resistance to *M. oryzae* in rice [39,40] within the significant QTLs *qSPAD12* of SPAD value at the seedling stage. At chromosome 6, another functional gene *LOC_Os06g12300* was observed encoding an expressed protein, drought-induced proteins, and anther and pollen wall remodeling/metabolism proteins, which contribute to the tolerance of rice to salt stress between the chromosomal region 6,643,235–6,643,552 bp inside the QTL *qNa6*; here, the total number of the loci were 11. The QTLs of the K^+^ conc. (*qK8*) and Na^+^/K^+^ ratio (*qNaK-R8*) were found in chromosome 8. However, their QTL position and the total numbers of loci, candidate genes, and putative functions were different from one another. The QTL position and the total numbers of loci, candidate genes, and putative functions of K^+^ conc. (*qK8*) were 4,794,164–4,799,199 bp, 12, *LOC_Os08g08350*. Retrotransposon proteins and the Ty1-copia subclass expressed tuning gene expression during plant development salinity and played a major role in shaping genome structure. The QTL position, total numbers of loci, candidate genes, and putative functions of Na^+^/K^+^ ratio (*qNaK-R8*) were 4,333,717–4,335,434 bp, 13, *LOC_Os08g07740,* and the histone-like transcription factor and archaeal histone regulated vegetative growth, sexual reproduction, virulence, and hyperosmotic stresses in response to salt stress. The QTL of K^+^ conc. (*qK12*), found in the region 18,717,286–18,718,979 bp at chromosome 12 with 10 loci has putative functions, including transposon protein, CACTA, En/Spm sub-class, and expressed; it significantly contributes to genome size, producing a large number of cDNA sequences in plant tissues under different conditions of stress. In the chromosomal region 27,449,823–27,452,792 bp of chromosome 11, one functional gene *LOC_Os11g45380* was identified in the zinc finger family of proteins, promoting plant growth, development, and stress signal transduction, and played an effective role in stress tolerance within the significant QTL *qNaK-R11* of Na^+^/K^+^ ratio at the seedling stage.

### 2.8. Epistasis Interaction

Epistasis is critical for regulating quantitative characters by retaining interactions between alleles at several loci. For all traits, a two-way test was used to detect different types of interactions, namely interactions between complementary loci, between QTLs, and between QTLs and background loci, among alleles at multiple loci with a minimum LOD of 5.0 using the ICIM-EPI method from IciMapping version 4.2.0 software. The results of this analysis showed that there were 43 significant interactions. Here, two types of digenic interactions were identified: (i) complementary type; (ii) between-QTL and background. Three interactions for SES score, five marker loci intervals (MI) for survival rate, five MI for shoot length, five MI for shoot dry weight, two interactions for root length, eight epistasis interactions for Na^+^, and four epistasis interactions for K^+^ concentration were found. There were also eight intervals for the SPAD value and three intervals for the Na^+^/K^+^ ratio (Table 4). There were two types of digenic interactions that were found: between complementary loci and between-QTLs background loci (Table 4; Figure 5). *qSUR5.2,* which is on chromosome 5, interacts with background loci (such as marker interval C763-C764), which have a LOD of 10 for survival rate (Table 4; Figure 5c). *qSUR5.2* also interacts with complementary locus 3, which is on chromosome 3 and has a LOD of 75.8 for K^+^ concentration (Table 4; Figure 5c). These two loci have a LOD of 10 for the survival rate (Figure 5b). Background loci at MI: C38-C39 (55.8 cM) on chromosome 1 and MI: C417-C418 on chromosome 5 (50.02 cM) interact with each other. The LOD for Na^+^ concentration was 6 and the PVE for Na^+^ concentration was 10.06. (Table 4; Figure 5a). This study does not show that QTLs interact with each other, demonstrating that there are strong interactions between the QTL and background loci, as well as 41 interactions between complementary loci. Two QTL and background epistasis interactions had a high LOD value (10) and a PVE of 2.22 percent on chromosome 10. Additionally, out of 41 complementary loci, the root length has a high LOD value (89) with a PVE value of 7.64%.

## 3. Discussion

### 3.1. Salt Stress Responses of the Parental Lines and Selected F_2:3_ Progenies and Path Coefficient Analysis

The main traits are SES, Na^+^ concentration, K^+^ concentration, Na^+^/K^+^ ratio, survival rate, SPAD value, shoot and root length and shoot biomass, which all play an important role in salt tolerance. In this investigation, the tolerant parent SES scores ranged from 4 to 5 with an average of 4.5, but the insensitive parent SES score was from 6 to 7 with an average of 6.5. In the sensitive parent, the value of Na^+^ concentration was more than 0.10 mmolg^−1^dwt, whereas the tolerant parent had 0.07 mmolg^−1^dwt and a high Na^+^/K^+^ ratio for salt-sensitive individuals, which was lower in salt-tolerant individuals. Water absorption is lower when Na^+^ accumulation is higher; as a result, plant growth was extremely hampered since the enzymatic connection was disrupted in the plant cell, which also disrupted the proteins and ultimately caused the early death of the plant, reducing grain yield [8,47,48,49,50,51]. The tolerant set had a high survival (95%), but the sensitive set had a lower survival (50%). In this study, the tolerant set had a high shoot length of 55.0 cm, ranging from 53.6 cm–56.4 cm, but the sensitive set had a lower shoot length of 28.85 cm. The tolerant individual had a higher shoot biomass (0.22 g), and the sensitive individual had a lower biomass (0.05 g), strongly supporting a larger amount of biomass and longer shoot length, which enhanced faster growth at the seedling stage where the Na^+^ concentration was diluted in the plant tissue/body [8,50,51,52].

There was a negative correlation among the SES score with K^+^ concentration, survival root length, and SPAD value; shoot length with SPAD value; and Na^+^/K^+^ ratio with shoot length, K^+^ concentration, SPAD value, and root length. In saline conditions, a higher amount of Na^+^ concentration accumulates in the plant body, causing the early death of the plant and reducing the grain yield [8,47,48,49,50,51]. The relationship between the different variables helps us to select procedures in the plant breeding program. The strong and positive relationship between SES score and Na^+^ concentration was shown to have an impact on the total phenotypic responses, as previously reported in [8,53,54].

Different independent traits, such as survival, shoot length, shoot dry weight, root length, SPAD value, Na^+^ concentration, K^+^ concentration, and Na^+^/K^+^ ratio, directly and indirectly contributed to the salt tolerance (dependent trait). Thus, the total salt tolerance is the sum of these eight characteristics and includes a residual effect (R), which is consistent with earlier research [55,56,57,58,59]. A path coefficient analysis revealed that survival, Na^+^ concentration, K^+^ concentration, and Na^+^/K^+^ ratio are the most crucial features for estimating the degree of salt tolerance, as these variables collectively account for the majority of phenotypic variance. Several researchers also made similar observations [8,14,50,60,61,62,63]. We need to select the advanced line based on these four characteristics (survival, Na^+^ concentration, K^+^ concentration, and Na^+^/K^+^ ratio) for a successful breeding program of salt-tolerant variety development and QTL dissection. Thus, path coefficients can play a vital role in developing salinity-tolerant varieties through trait prioritization and selection.

### 3.2. Genetic Map Construction and QTL Detection

Previously, genetic maps were constructed, mainly using low-throughput marker systems, such as simple sequence repeats (SSR), amplified fragment length polymorphisms (AFLP), restriction fragment length polymorphisms (RFLP), and random amplified polymorphic DNA (RAPD) [64]. The maps usually comprised less than 300 markers with more than 30 cm gaps found between markers in some chromosomal regions [65,66]. Therefore, high-throughput SNP marker technology with a multiplexing ability is currently used to construct high-resolution and dense genetic linkage maps. The linkage map constructed in this study revealed an average interval of 1.7 cm between marker loci demonstrating an acceptable saturation with SNP markers for potential applications in QTL discovery.

Most of the studies on salinity tolerance at the seedling stage were carried out using a few common donors, such as Pokkali, Nona Bokra, and Capsule. However, we made a few attempts to identify a new salt tolerance donor, novel QTL, and identify epistasis interactions for salinity tolerance. Akundi, a Bangladeshi landrace used as a donor in this study, shows a higher level of salt tolerance in the hotspot region in the coastal areas of Bangladesh [14]. Here, we also used BRRI dhan49, which is sensitive to salt but a very popular *Aman* variety in Bangladesh. The major findings are that a total of 15 QTLs were reported on seven chromosomes. Multiple QTLs were identified on chromosome 1 (*qSES1*, *qSUR1*, *qSL1*, *qRL1*), chromosome 3 (*qSES3*), chromosome 5 (*qSUR5.1*, *qSUR5.2*, *qSDW5*), chromosome 6 (*qNa6*), chromosome 8 (*qK8*, *qNaKR8*), chromosome 11 (*qSDW11*, *qNaKR11*), and chromosome 12 (*qSPAD12*, *qK12*) using the QGene and ICIM-ADD software. Fifteen candidate genes and their approximate positions were identified, and a number of loci and putative functions also corresponding to these QTLs. Additionally, 43 significant epistasis interactions were detected in this study using ICIM-EPI software. These epistatic/digenic interactions are important for controlling quantitative traits through interactions between alleles at several loci such as between QTL vs. QTL, between QTL vs. background loci, and between background loci.

### 3.3. Genomic Regions for Salinity Tolerance

A total of 15 OTLs identified in the present study, which were associated with agronomic and physiological parameters. On chromosome 1, four QTLs, *qSES1*, *qSL1*, *qSUR1*, and *qRL1*, characterized nearly 82.0% of the overall phenotypic variation. Further investigations should be focused on gaining a better understanding of their critical functions in altering salt tolerance during physiological processes [8].

In the present study, five common QTLs (*qSUR1*, *qSUR5.2*, *qSLqN1*, *qSDW10*, *qNa10*, and *qK1*) were identified both in QGene and ICIM-ADD software with higher PVE (>10.0%); thus, these QTLs are considered to be major QTLs.

In a multi-locus analysis, a combined effect of some major QTLs *qSES1* and *qSL1,* was found on chromosome 1 (combined PVE 46.3% of two QTLs with R^2^ 15.6% and 30.7%, respectively), as well as *qSUR1* and *qRL1* on chromosome 1 (combined PVE 35.7% with R^2^ 19.9% and 15.8%, respectively). When the major QTLs of these characters were measured concurrently, their overall summation of individual effect was higher than their combined effect. These are because of (i) the co-localization of the QTLs, (ii) the high additive epistasis interaction among QTLs, and (iii) the use of similar pathways by some of the QTL affecting this trait.

### 3.4. Comparison between New QTL from the Current and Previously Mapped QTLs

The QTLs detected in this experiment were compared to those previously recognized in various studies. In the vegetative stage, a major salt-tolerant QTL was identified as *Saltol* QTL, which was mapped in chromosome 1 and had a strong phenotypic effect (R^2^ of 39% to 44%) from the landrace Pokkali [19,67,68,69,70]. Another major salt-tolerant gene is *SKC1,* also identified in chromosome 1 during the vegetative stage from the Nona Bokra [8,71,72,73]. These two genes, *Saltol* (10.5–11.5 Mb) and *SKC1* (11.46 Mb), were absent in Akundi. Two completely new QTLs (*qSES1* and *qSL1*) on chromosome 1, (position: 151.8–156.0 cM) which are different from previously reported QTL *Saltol* [19], and two QTLs (*qSUR1*, *qRL1*) were identified on chromosome 1, and their position (48.8–50.0 cM) was co-located. Therefore, these new QTLs might show a higher level of tolerance to salt stress.

The QTLs and their position reported in this study are different from other studies [8,50,51,74,75,76] except for *qSL1*. Only *qSL1* QTL shared the same map location. Therefore, from this comparative analysis, we strongly suggest that the remaining 14 QTLs are novel biomarkers for use in rice breeding to enhance salt tolerance.

Several QTLs had a significant effect on chromosome 1. One QTL for *qSES1* had a LOD of 3.4 and R^2^ of 15.6%; the other QTLs were *qSL1* (LOD = 7.2, R^2^ = 30.7%), *qSUR1* (LOD = 4.3, R^2^ = 19.9%) and *qRL1* (LOD = 3.3, R^2^ = 15.8%). These two sets of QTLs are co-localized and have functional relatedness. Therefore, these major QTLs have pleiotropic effects on other traits. Co-localized QTL on chromosome 1 for different traits were also strongly correlated among the traits.

The four major QTLs, such as *qSES1*, *qSL1*, *qSUR1*, and *qRL1*, were found in this study and could be used for pyramiding and marker-assisted breeding. Here, a total of 14 QTLs were identified as novel QTLs that could be used in future breeding programs to improve salt tolerance.

## 4. Materials and Methods

### 4.1. Parent Selection

BRRI dhan49 and Akundi were selected as parents to develop the mapping population. Akundi is an *indica* landrace that is salt-tolerant at the early seedling stage and was grown in the southern coastal region of Bangladesh. It grows to a height of 150–155 cm; is photoperiod-sensitive, produces strong seedlings; has long, wide, and droopy leaves, around 6–7 tillers; and its panicles are 15–20 cm long. Usually, grains are awnless and reddish in appearance with a lower number of grains. They have a low yield potential (2.0–2.5 t ha^−1^) and mature in 140–145 days [14].

BRRI dhan49 is a common *indica* rainfed lowland rice type with a moderate height (100 cm), awnless, and with medium slender grains; it matures in 135 days and yields 5.5 t ha^−1^. It is susceptible to salt stress at the vegetative stage but moderately responsive to photoperiods [77].

### 4.2. Details of the Cross, Confirmation and Management of the F_1_s and the Segregating Population

Two parents, including a salt-tolerant donor (Akundi) and a recipient (BRRI dhan49), were grown on four different sowing dates for hybridization. Seeding started in June 2019 with an interval of seven days to synchronize flowering times and achieve desired cross combinations. The F_1_ seeds, along with their respective parents, were germinated in the petridish. Thirty-day-old seedlings were transplanted in a 5.4 m × 2 rows plot with a spacing of 20 × 15 cm in the Aman season. A single seedling was used for transplanting. Growing conditions including field and phytotron management practices are shown below (in 4.3). Hand weeding was carried out in time. Plant protection measures were taken for disease and insect infestation if necessary. Leaf samples and respective parents were collected from each of the F_1_ plants for quality-check (QC) genotyping to determine true F_1_s. QC genotyping was performed using 10 QC SNPs at Intertek, Australia. The cross was confirmed through F_1_ verification using quality-check (QC) genotyping with purity SNP panel, as well as careful observation of plant characters in the Aman season. After confirmation, the F_2_ seeds were collected. This F_2_ population-derived F_2:3_ family was used as the mapping population for salt tolerance, which was obtained from a cross between BRRI dhan49 and Akundi; it was used in the present investigation to map salt-tolerant QTLs.

### 4.3. Growing Conditions

This study was performed at the Bangladesh Rice Research Institute (BRRI; http://www.brri.gov.bd (accessed on 22 April 2022), Gazipur, Bangladesh, during the Aman season. Two parents were grown in three sets beginning on 20 June 2019, with a seven-day interval between each set to synchronize flowering times for making the desired cross. A single twenty-day-old seedling was transplanted in a plot of 5.4 m × 4 rows with 20 × 20 cm spacing in hybridization block. Fertilizer doses were 90, 60, 40, and 20 kg/ha N, P, K, and S, respectively. Nitrogen was applied in three equal splits at 15 days after transplanting. At the time of final land preparation, the entire quantity of P, K, and S was applied. Intercultural operations were carried out as needed. Then, two crosses were made in the Plant Breeding Division’s net house. To disinfect the F_1_ seeds and their parents, 0.1 percent mercuric chloride solution was used. Seeds were germinated on Petri dishes and sown into earthen pots using treated seeds. The seedling was transplanted in 5.4 m single-row plots at a spacing of 20 cm × 20 cm along with their respective parents with proper labeling. Then, F_1_ and F_2_ seeds were obtained and saved from the selected crosses. The following rainfed season, in June 2020, F_2_ seeds were grown, and a single seedling was transplanted. Then, leaf samples were collected 15 days after transplanting for genotyping. In hydroponics, the F_2:3_ individuals were phenotyped in a phytotron set at 30 °C/22 °C day/ night temperature and 65–70% relative humidity in 2021. Two pre-germinated seeds were sown per hole in 10 L plastic trays using styrofoam seedling floats floating in distilled water for 3 days, followed by the use of a culture solution [78]. Salt stress was induced 14 days after seeding by adding NaCl to the culture solution until an electrical conductivity of 12 dS m^−1^ was achieved. To avoid lodging, silicon in the form of sodium metasilicate 9 hydrate (4.5 mg L^−1^) was added. To avoid Fe deficiency, the nutrient solution was acidified daily to a pH of 5.0, and the solution was replaced every 7 days.

### 4.4. Characterization of Agronomic Traits

Ninety-one (91) F_2:3_ individuals were characterized based on specific morphological and physiological characteristics. Seedling salt stress injury symptoms were phenotypically evaluated using SES scores (Standard Evaluation System; [79]), with a value of 1 indicating highly tolerant genotypes and 9 representing highly sensitive crops. Individual plants were taken 21 days after salinization, dried for 3 days at 70 °C, and then weighed. Sodium–potassium levels were then determined [14,80,81]. Three weeks following salinization, seedling survivors were recorded, and survival % was computed. The shoot length was determined by measuring the distance between the base of the stem and the top of the highest leaf. Each plant was harvested from the root to all plant components and oven-dried at 70 °C for 3 days before being weighed. The root length was measured from the stem base to the tip of the tallest root.

### 4.5. Physiological Characterization

Each plant was picked from a different F_2:3_ family and rinsed three times with deionized water before drying. The individual plant was then washed and dried in the sun for 3 days before being dried at 50 °C for 3 days. After drying, the material was crushed and weighted to approximately 0.50 g before being placed in a test tube with 25 mL 1N HCL. After 24 h of digestion in 1N HCL, the samples were filtered, and the extract was extracted from the test tube using filter paper. The extract was then diluted with 39 mL 1N HCL and 1 mL liquid phase of the extract. Then, a reference solution was produced, and the sodium and potassium concentrations were determined using a flame photometer (Model410) [78].

### 4.6. SPAD Reading

Five completely expanded second leaves were chosen for each replication to determine the chlorophyll concentration. The chlorophyll content of leaves was determined (non-destructive method) using a SPAD meter (Konica Minolta 502, Tokyo Japan) just prior to harvesting.

### 4.7. Correlation Analysis for Trait Associations

A set of 91 F_2:3_ progenies were selected for additional study from one cross of BRRI dhan49/Akundi. The correlation coefficients involving various traits were evaluated using the software RStudio 4.1.1.

### 4.8. Path Coefficient Analysis

Path coefficient analysis was applied to estimate the direct and indirect contributions of various attributes (survival rate, shoot length, shoot dry weight, root length, SPAD value, Na^+^ concentration, K^+^ concentration, and Na^+^/K^+^ ratio) to overall salt tolerance score (SES: standard evaluation score based on morphological/phenotypic manifestation of salt stress symptoms) [82,83].

### 4.9. Residual Effect

It is very difficult to obtain a full understanding of all the component traits in plant breeding. The residual effect shows detailed explanations of how other possible variables, which are not studied, interact. In other words, the residual effect quantifies the effect of other possible independent factors on the dependent variable that were not included in the study. The direct effect and simple correlation coefficients were used to calculate how much of an effect there was leftover.

### 4.10. Genotyping and Construction of a Genetic Linkage Map

Leaf samples from 91 individuals of an F_2_ population derived from the BRRI dhan49/Akundi cross were collected from young leaves of 3-week-old plants and then stored at an ultra-low temperature of −80 °C for 1k-RiCA genotyping (a powerful customized sequencing-based amplicon panel consisting of ~ 1000-SNPs; [84]). During sample preparation, each sample was punched into a small piece and inserted into a specific well of 96-well plates according to sample number. Then, the sample plates were oven-dried at 50 °C for 24 h and wrapped in a zipper bag. Finally, the samples were oven-dried for 1k-RiCA panel (SNP marker test). Whole-genome genotyping for the 91 rice genotypes was carried out using genotyping by sequencing technique using 950 SNP markers. Genotyping was carried out at AgriPlex Genomics, 11000 Cedar Avenue, Suite 250. Cleveland, OH 44106, USA, who was a service provider. Molecular maps were constructed using the 950 SNP markers, and the marker distances were computed by multiplying the Mb positions by 3.924 to achieve an equivalent estimate in cM.

### 4.11. QTL Analysis

Marker analysis and QTL mapping were performed on the 91 F_2_ plants selected for BRRI dhan49/Akundi. These progenies were genotyped using SNP markers (Appendix A), and 884 bin markers covering the BRRI dhan49/Akundi rice genome were utilized to generate a molecular linkage map. The genetic linkage map was dispersed by 1.7 cM on average. To determine the correlation between specific marker loci and salinity-tolerance-related phenotypes, a QTL analysis was conducted using QGene 4.0 program [85]. Interval mapping (IM), composite interval mapping (CIM), and single-marker regression (SMR) analyses were used to pinpoint the position of the discovered QTL for salt tolerance. The least LOD value expected to identify a QTL significant was determined using 1000 permutations runs [86]. R^2^ value was computed as the fraction of total variation described by each QTL (R^2^ = PVE, phenotypic variation explained by the QTL). For the CIM, forward cofactor selection method commands were employed. For each trait, the fraction of phenotypic variance (R^2^ value) and additive effects were calculated.

## 5. Conclusions

The leading traits for salinity tolerance are SES, Na^+^ concentration, K^+^ concentration, Na^+^/K^+^ ratio, survival rate, shoot and root length and shoot biomass, which all play an important role in salt tolerance. We used trait association and a path analysis to assess the contribution of the different salt-tolerance-related traits at the early vegetative stage in the rice landrace, Akundi. The results suggest that if Na^+^ concentration and Na^+^/K^+^ ratio increase in plant tissue, this can cause early damage to rice plants. Thus survival rate, Na^+^ concentration, K^+^ concentration, Na^+^/K^+^ ratio are the key pathways at the seedling stage for salt tolerance in rice. A total of 15 new QTLs were identified through 1k-RiCA SNP assay using single markers regression, interval mapping, and composite interval mapping method. The QTLs identified on chromosome 1 (*qSES1*, *qSUR1*, *qRL1*); 3(*qSES3*); 5(*qSUR5.1*, *qSUR5.2*, *qSDW5*); 6(*qNa6*); 8(*qK8*, *qNaKR8*); 11(*qSDW11*, *qNaKR11*); 12(*qSPAD12*, *qK12*). Out of these 15 QTLs, 14 QTLs are novel because they do not match the position of other previously reported QTLs. In this study, four major QTLs (*qSES1*, *qSL1*, *qSUR1,* and *qRL1*) were identified and these four QTLs had a major effect on salt tolerance. Here, we also identified 15 candidate genes (*LOC_Os01g66670*, *LOC_Os03g05770*, *LOC_Os01g22700*, *LOC_Os05g24680*, *LOC_Os05g05230*, *LOC_Os01g68490*, *LOC_Os05g35310*, *LOC_Os11g18550*, *LOC_Os01g22150*, *LOC_Os12g42440*, *LOC_Os06g12300*, *LOC_Os08g08350*, *LOC_Os12g31120*, *LOC_Os08g07740*, *LOC_Os11g45380*) and 43 significant epistasis interactions that spread across 12 different chromosomes. These QTLs might be useful for developing highly salt-tolerant rice varieties using QTL pyramiding and marker-enabled breeding for salt-affected coastal areas. These stringent and robust QTLs and interacting QTLs may be potential targets for detailed research through fine mapping and map-based cloning for salt tolerance.

## Figures and Tables

**Figure 1 plants-11-01409-f001:**
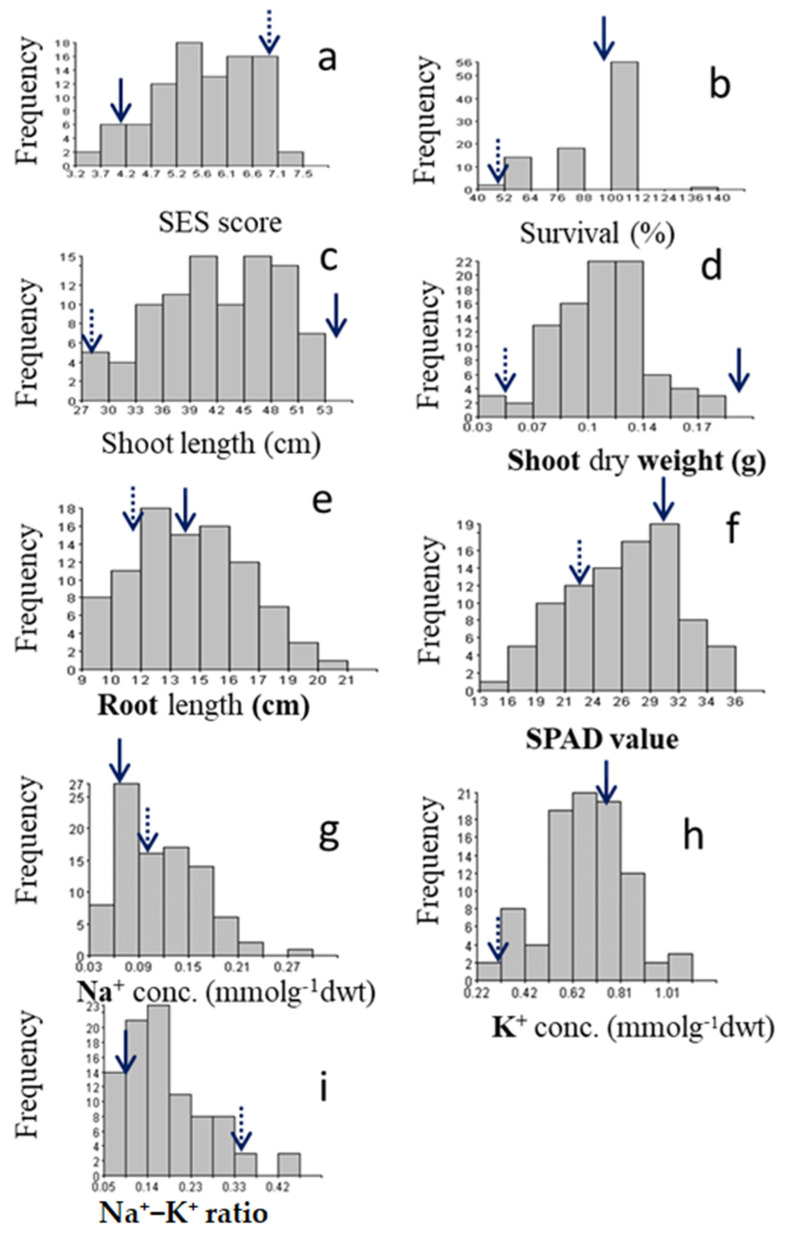
Histogram illustrates the distribution (number of plants) of the F_2:3_ individuals for traits associated with salinity tolerance at the seedling stage: (**a**) SES score, (**b**) survival rate, (**c**) shoot length, (**d**) shoot dry weight, (**e**) root length, (**f**) SPAD value, (**g**) Na conc. (**h**) K conc. (**i**) Na/K ratio. Solid and dotted arrows indicate the phenotypic value of the tolerant parent Akundi and sensitive BR49, respectively.

**Figure 2 plants-11-01409-f002:**
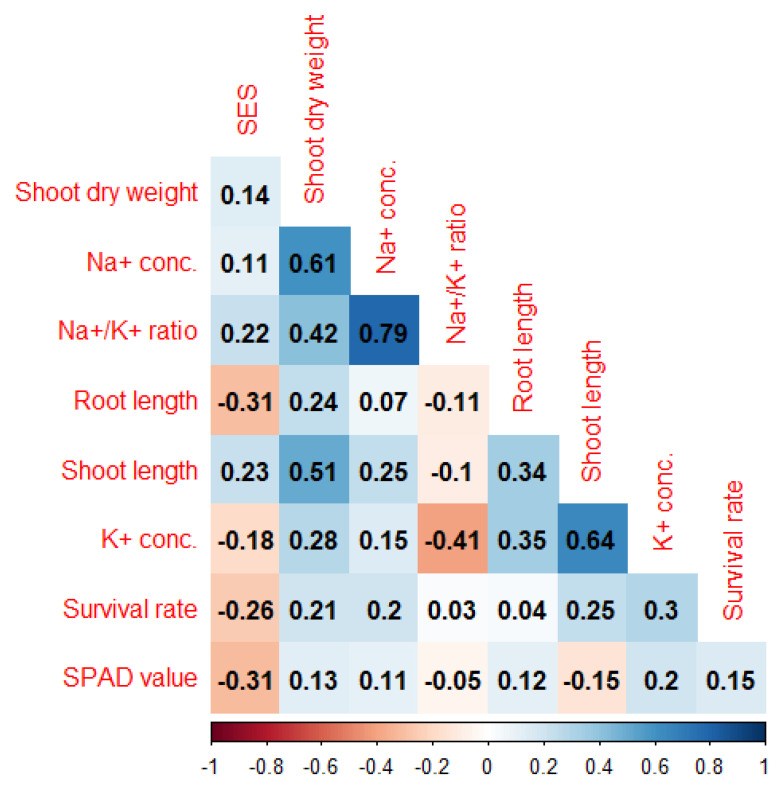
Correlation coefficients among different traits related to salt-tolerance in the F_2:3_ population of a cross between BR49 (salt-sensitive) and Akundi (salt-tolerant) at seedling stage. Conc.: Concentration. Tabulated *t*-value at 5% level = 0.205, at 1% level = 0.267 indicate significance at *p* < 0.05 and *p* < 0.01, respectively.

**Figure 3 plants-11-01409-f003:**
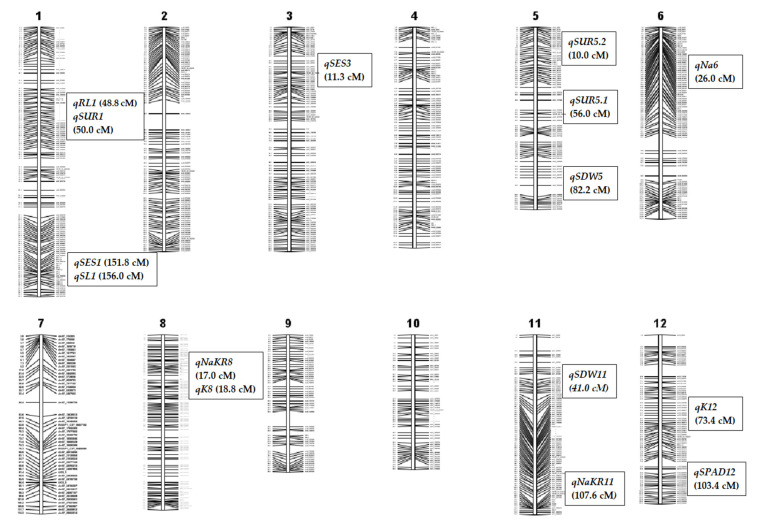
A genetic linkage map of 12 chromosomes was constructed using 884 SNP markers. This genetic map shows the distribution of detected QTLs on seven different chromosomes of rice. The map was built based on selected individuals of an F_2:3_ population of a cross between BR49 and Akundi. The number at the top of each linkage group indicates the chromosome number. The names of the SNP markers are shown on the right, and the rectangular boxes beside SNPs represent the approximate locations of the QTL identified for salt tolerance.

**Figure 4 plants-11-01409-f004:**
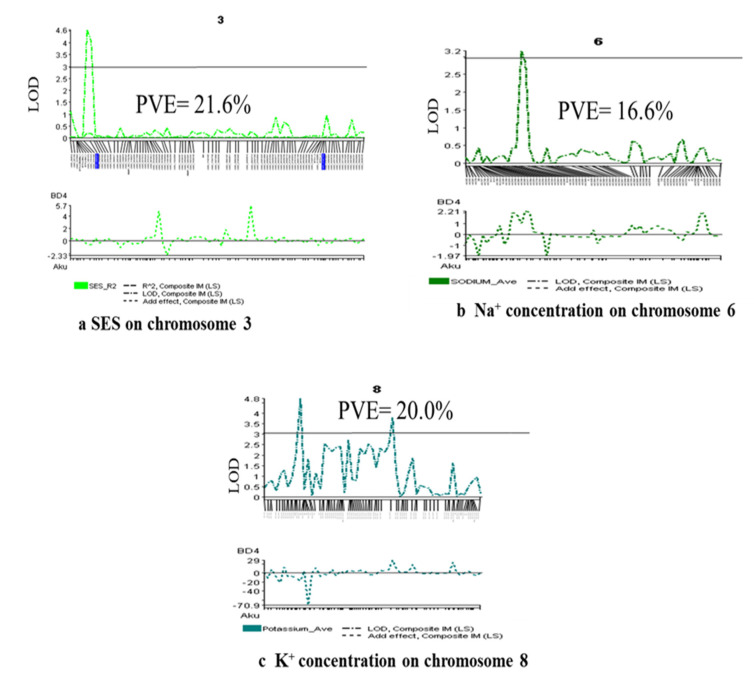
QTL likelihood curves showing the LOD score of (**a**) SES score on chromosome 3, (**b**) Na^+^ concentration on chromosome 6 and (**c**) K^+^ concentration on chromosome 8 were above the significance threshold of LOD = 3.0 and explained approximately 21.6%, 16.6% and 20.0% of the phenotypic variance for each trait.

**Figure 5 plants-11-01409-f005:**
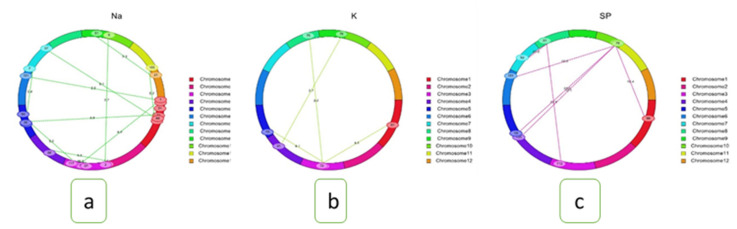
Cyclic diagrams of epistatic interactions of QTLs (E-QTLs) associated with salt tolerance indices: (**a**) Na^+^ concentration, (**b**) K^+^ concentration, and (**c**) survival rate. The 12 colors in the ring indicate the 12 chromosomes of rice. The numbers in the ovals represent the marker positions on chromosomes. The dotted lines indicate the interacting marker pairs located on the same or different chromosomes with corresponding LOD scores due to epistatic effects (E-QTLs).

**Table 1 plants-11-01409-t001:** Correlation and path coefficients for direct and indirect effects of survival rate, shoot length, shoot dry weight, root length, SPAD value, Na^+^ concentration, K^+^ concentration, Na^+^/K^+^ ratio on SES scores in an F_2:3_ population of a cross between Akundi (salt-tolerant) and BR49 (salt-sensitive) grown under salt stress at seedling stage.

Variable	Correlation	Survival Rate	Shoot Length	Shoot Dry Weight	Root Length	SPAD Value	Na^+^ Conc.	K^+^ Conc.	Na^+^/K^+^ Ratio	Total Effect
Survival rate	−0.257	**−0.301**	0.148	0.009	−0.014	−0.011	−0.003	−0.088	0.002	−0.257
Shoot length	0.226	−0.074	**0.605**	0.024	−0.134	0.010	−0.004	−0.190	−0.011	0.226
Shoot dry weight	0.141	−0.059	0.304	**0.047**	−0.096	−0.008	−0.010	−0.083	0.046	0.141
Root length	−0.306	−0.011	0.208	0.012	**−0.390**	−0.008	−0.001	−0.103	−0.012	−0.306
SPAD value	−0.310	−0.046	−0.089	0.006	−0.046	**−0.068**	−0.002	−0.058	−0.005	−0.310
Na^+^ conc.	0.114	−0.060	0.151	0.028	−0.026	−0.007	**−0.016**	−0.044	0.088	0.114
K^+^ conc.	−0.183	−0.090	0.388	0.013	−0.136	−0.013	−0.002	**−0.296**	−0.045	−0.183
Na^+^/K^+^ ratio	0.218	−0.007	−0.060	0.019	0.042	0.003	−0.013	0.120	**0.112**	0.218

Residual effect: 0.44. The numbers written in bold indicate the direct effect.

**Table 2 plants-11-01409-t002:** Locations and features of QTLs detected for seedling-stage salt-tolerance-related agronomic and physiological traits.

Trait	QTL	Chr.	Peak Marker	QTL Position (cM)		Add Effect	LOD		PVE (%)		Method of QTL Detection	Contributor of Favorable Allele
				QGene	ICIM		QGene	ICIM	QGene	ICIM		
SES	*qSES1*	1	Chr01_38632196	151.8	-	−0.61	3.4	-	15.6	3.80	SMR	Akundi
	*qSES3*	3	Chr03_2885423	11.3	-	−0.70	4.8	-	21.6	4.50	SMR, CIM	Akundi
Survival rate (%)	*qSUR1*	1	chr01_12941687	50	54.8	31.91	4.3	5.0	19.9	0.98	IM, CIM	BR49
	*qSUR5.1*	5	chr05_14412355	56	-	20.25	7.5	-	31.6	-	IM, CIM	BR49
	*qSUR5.2*	5	chr05_2291156	10	11.2	−10.36	3.1	5.0	14.9	0.94	IM, CIM	Akundi
Shoot length	*qSL1*	1	QSES1-2_2	156	151.8	4.86	7.2	4.0	30.7	16.28	IM, CIM	BR49
Shoot dry weight	*qSDW 5*	5	chr05_20551103	82.2	-	0.35	3.5	-	16.3	-	IM, CIM	BR49
	*qSDW11*	11	Chr11_10741559	41.0	-	0.27	3.0	-	14	-	IM, CIM	BR49
Root length	*qRL1*	1	Chr01_12335190	48.8	-	−1.59	3.3	-	15.8	-	IM	Akundi
SPAD	*qSPAD12*	12	Chr12_26259494	103.4	-	3.05	3.5	-	16.3	-	IM, CIM	BR49
Na^+^ conc.	*qNa6*	6	SSIIA-3B	26	27	−1.98	3.6	4.0	16.8	8.18	CIM, IM	Akundi
K^+^ conc.	*qK8*	8	chr08_4853081	18.8	-	−0.34	4.7	-	21.2	-	IM, CIM	Akundi
	*qK12*	12	Chr12_18881059	73.4	-	11.4	3.5	-	16.3	-	IM, CIM,	BR49
Na^+^/K^+^ ratio	*qNaKR8*	8	DTH8-IR24	17	-	0.08	3.0	-	0.1	-	SMR	BR49
	*qNaKR11*	11	IRGSP1_C11_27391141	107.6	108	−0.04	3.0	3.0	12.9	16.80	SMR`	Akundi

Chr.: chromosome, Add effect: additive effect, SMR: single marker regression, IM: interval mapping, CIM: composite interval mapping, PVE: phenotypic variation explained, ICIM: inclusive composite interval mapping.

**Table 3 plants-11-01409-t003:** Possible candidate gene loci within the QTL regions illustrating putative functions and references.

Traits	QTL	Chr.	Position (bp)	Total no. of Locus	Candidate Genes	Putative Function	References
SES	*qSES1*	1	38,723,347–38,724,165	16	*LOC_Os01g66670*	Expressed protein (drought-induced proteins, anther and pollen wall remodeling/metabolism proteins contribute to the tolerance of rice to salt stress).	[25,26]
	*qSES3*	3	2,878,828–2,880,890	20	*LOC_Os03g05770*	Peroxidase precursor, putative, expressed (increases protection against oxidative stress and is highly tolerant to different stresses, allowing survival when water supply is a limiting factor).	[27,28]
Survival rate (%)	*qSUR1*	1	12,756,935–12,757,588	17	*LOC_Os01g22700*	Organic cation transporter-related, putative, expressed (affects root development, carnitine-related responses under stress, and numerous biological processes, including transcription, translation, cell signaling, and ion channel activity).	[29,30]
	*qSUR5.1*	5	14,280,156–14,292,389	10	*LOC_Os05g24680*	Retrotransposon protein, putative, Ty3-gypsy subclass, expressed (enable plants to cope with drought stress conditions; impact on biological process under stress response).	[31,32]
	*qSUR5.2*	5	2,548,592–2,559,171	17	*LOC_Os05g05230*	Expressed protein (drought-induced proteins, anther and pollen wall remodeling/metabolism proteins contribute to the tolerance of rice to salt stress).	[25,26]
Shoot length	*qSL1*	1	39,794,226–39,799,341	18	*LOC_Os01g68490*	Tetratricopeptide-like helical, putative, expressed (abscisic acid responses and osmotic stress tolerance enable plants to cope with adverse environmental conditions).	[33,34]
Shoot dry weight	*qSDW 5*	5	20,966,622–20,969,373	17	*LOC_Os05g35310*	Ankyrin repeat family protein, putative, expressed (implicated in plant growth, development and signal transduction; response to biotic and abiotic stresses).	[35,36]
	*qSDW11*	11	10,456,526–10,459,838	18	*LOC_Os11g18550*	Transposon protein, putative, CACTA, En/Spm sub-class, expressed (contributes significantly to genome size, produces a large number of cDNA sequences in plant tissues different conditions of stress).	[37,38]
Root length	*qRL1*	1	12,444,630–12,446,150	12	*LOC_Os01g22150*	Transposon protein, putative, CACTA, En/Spm sub-class (contributes significantly to genome size, produces a large number of cDNA sequences in plant tissues under different conditions of stress).	[37,38]
SPAD	*qSPAD12*	12	26,377,868–26,379,849	17	*LOC_Os12g42440*	Chaperone protein dnaJ, putative, expressed (response to NaCl stress, involved in basal resistance to *M. oryzae* in rice).	[39,40]
Na^+^ Conc.	*qNa6*	6	6,643,235–6,643,552	11	*LOC_Os06g12300*	Expressed protein (drought-induced proteins, anther and pollen wall remodeling/metabolism proteins contribute to the tolerance of rice to salt stress).	[25,26]
K^+^ Conc.	*qK8*	8	4,794,164–4,799,199	12	*LOC_Os08g08350*	Retrotransposon protein, putative, Ty1-copia subclass, expressed (tuning gene expression during plant development salinity; plays a major role in shaping genome structure and size during salinity).	[41,42]
	*qK12*	12	18,717,286–18,718,979	10	*LOC_Os12g31120*	Transposon protein, putative, CACTA, En/Spm sub-class, expressed (contributes significantly to genome size, produces a large number of cDNA sequences in plant tissues different conditions of stress).	[37,38]
NaK ratio	*qNaK-R8*	8	4,333,717–4,335,434	13	*LOC_Os08g07740*	Histone-like transcription factor and archaeal histone, putative, expressed (regulating vegetative growth, sexual reproduction, virulence and hyperosmotic stresses, response to salt stress).	[43,44]
	*qNaK-R11*	11	27,449,823–27,452,792	12	*LOC_Os11g45380*	Zinc finger family protein, putative, expressed (plant growth, development, and stress signal transduction, effective role in stress tolerance).	[45,46]

**Table 4 plants-11-01409-t004:** Epistatic interaction describes how epistatic QTL interactions can affect the traits related to salinity tolerance at the seedling stage.

Traits	Chr1	Position1 (cM)	FM1	Chr2	Position2 (cM)	FM2	LOD	PVE (%)	Add1	Add2	Add by Add	Type of interaction
SES	4	10.7	C308–C309	7	27	C548–C549	42	2.628	−0.003	−0.953	−0.951	Between complementary loci
	2	120.4	C190–C191	12	26.4	C865–C866	48	2.637	−0.706	1.198	−0.206	Between complementary loci
	7	67	C557–C558	12	26.4	C865–C866	45	2.638	−0.322	0.322	−0.674	Between complementary loci
Survival	3	111.9	C276–C277	8	30.8	C608–C609	21	0.056	3.260	−0.444	−2.893	Between complementary loci
	4	130.7	C380–C381	10	70.6	C763–C764	10	1.826	12.950	4.923	−10.412	Between complementary loci
	5	10.2	C394–C395	10	70.6	C763–C764	10	1.947	4.950	−6.390	1.673	Between complementary loci
	6	101	C516–C517	10	70.6	C763–C764	10	1.584	3.374	2.100	−14.695	Between complementary loci
	1	65.8	C47–C48	10	75.6	C770–C771	10	2.222	1.557	7.981	6.373	Between QTLs and background
Shoot Length	2	15.4	C132–C134	2	75.4	C160–C161	6	9.206	4.177	0.113	−4.192	Between complementary loci
	2	75.4	C160–C161	8	70.8	C640–C641	6	6.744	1.225	−5.466	3.291	Between complementary loci
	8	15.8	C595–C596	8	70.8	C640–C641	5	9.325	−1.393	−1.560	6.540	Between complementary loci
	6	71	C507–C508	11	56	C799–C800	5	7.587	−2.065	−1.355	5.903	Between complementary loci
	7	7	C542–C543	11	96	C836–C839	5	7.724	0.024	0.512	3.333	Between complementary loci
Shoot dry weight	9	30.6	C694–C695	10	75.6	C770–C771	6	9.656	−0.001	0.000	−0.017	Between complementary loci
	2	75.4	C160–C161	11	21	C782–C783	6	9.723	−0.004	0.000	0.003	Between complementary loci
	12	11.4	C857–C858	12	56.4	C884–C885	8	8.903	−0.013	0.017	0.012	Between complementary loci
	6	86	C512–C513	12	76.4	C901–C902	19	1.480	0.003	0.004	0.007	Between complementary loci
	1	155.8	C104–C105	12	86.4	C906–C907	6	9.417	−0.003	0.011	−0.011	Between complementary loci
Root length	4	135.7	C381–C382	9	0.6	C671–C672	10	1.423	0.074	0.209	0.074	Between complementary loci
	8	75.8	C644–C645	12	76.4	C901–C902	89	7.637	−0.409	−0.427	−0.427	Between complementary loci
SPAD	1	115.8	C73–C74	3	136.9	C296–C297	6	8.485	−0.478	2.305	3.562	Between complementary loci
	1	115.8	C73–C74	4	55.7	C335–C336	5	5.614	2.343	4.534	1.190	Between complementary loci
	3	56.9	C246–C247	4	55.7	C335–C336	6	3.118	−2.093	0.707	4.291	Between complementary loci
	3	11.9	C217–C218	5	50.2	C417–C418	5	7.909	2.494	0.180	−1.053	Between complementary loci
	1	115.8	C73–C74	6	101	C516–C517	5	8.103	1.306	2.094	3.784	Between complementary loci
	4	25.7	C314–C315	7	27	C548–C549	6	6.784	−2.153	−0.795	1.959	Between complementary loci
	11	21	C782–C783	11	96	C836–C839	6	6.641	2.780	1.330	0.550	Between complementary loci
	9	45.6	C702–C703	12	81.4	C904–C905	6	1.772	−0.103	−0.503	0.583	Between complementary loci
Na^+^ Conc.	2	135.4	C199–C200	4	55.7	C335–C336	6	4.625	0.012	−0.028	−0.013	Between complementary loci
	3	116.9	C279–C280	5	45.2	C416–C417	5	0.533	0.006	0.006	−0.016	Between complementary loci
	1	55.8	C38–C39	5	50.2	C417–C418	6	10.06	−0.020	−0.022	0.019	Between complementary loci
	1	0.8	C1–C2	6	101	C516–C517	6	5.709	−0.015	0.012	−0.018	Between complementary loci
	1	65.8	C47–C48	7	97	C577–C578	6	8.641	−0.013	−0.046	0.037	Between complementary loci
	3	1.9	C208–C209	10	5.6	C728–C729	6	6.317	0.015	0.024	0.012	Between complementary loci
	1	65.8	C47–C48	11	106	C842–C843	5	9.005	0.019	−0.018	−0.049	Between complementary loci
	9	50.6	C706–C707	12	21.4	C865–C866	5	8.375	0.010	−0.011	−0.002	Between complementary loci
K^+^ conc.	1	90.8	C62–C63	3	66.9	C251–C252	9	12.65	−0.025	−0.014	0.014	Between complementary loci
	3	66.9	C251–C252	5	15.2	C397–C398	9	12.69	−0.014	−0.025	0.014	Between complementary loci
	3	71.9	C254–C255	8	75.8	C644–C645	9	1.718	0.002	−0.011	0.002	Between complementary loci
	4	90.7	C354–C355	9	75.6	C715–C716	6	8.798	0.026	0.000	0.000	Between complementary loci
Na^+^/K^+^ ratio	8	20.8	C599–C600	8	60.8	C637–C638	7	1.569	−0.016	−0.002	0.002	Between QTLs and background
	3	71.9	C254–C255	8	75.8	C644–C645	9	5.615	0.004	−0.017	0.004	Between complementary loci
	4	90.7	C354–C355	9	75.6	C715–C716	6	27.54	0.041	0.002	0.002	Between complementary loci

Chr1: Chromosome ID1; FM1: flanking markers (Position1); Chr2: chromosome ID2; FM2: flanking markers (Position2); LOD: LOD score due to epistatic effects; PVE (%): PVE due to epistatic effects (%); Add1: Estimated additive effect of position 1; Add2: Estimated additive effect of position 2; Add1 by Add2: Additive by additive epistatic at the two interacting positions.

## Data Availability

Not applicable.

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
