# Peer review of "Genetic Mapping to Detect Stringent QTLs Using 1k-RiCA SNP Genotyping Platform from the New Landrace Associated with Salt Tolerance at the Seedling Stage in Rice"

_plants, 2022, doi:10.3390/plants11111409_

Round 1

Reviewer 1 Report

Introduction first paragraph needs better organization and description

The Introduction and Discussion need editing

Make the citation format consistent:

…… were reported by [16]…. were reported by Wang et al. [16]

….. Mohammadi et al. [17]… correct

Reorganize the article by placing the sections as follows: abstract, introduction, materials and methods, results, discussion, and conclusions

Details of the cross and management of the F1 and the segregating populations are missing

Explain in detail what “under a salt environment” mean

What were the criteria to select for salt tolerance?

Explain the phenotypic effect on the seedlings or plants, by the salt injury

Indicate the various attributes evaluated by the Path coefficient analysis

Write the names of journals appropriately:

Advances in botanical research     Advances in Botanical Research

Nature communications             Nature Communications

Journal of genetics                      Journal of Genetics

The supplementary information would be better to be cited, since the file is not appropriate due its length, to be included in the article

Reviewer 2 Report

In this study, new QTL associated with salt stress tolerance were described in rice.

Please find below my suggestions.

Line 89: the salt concentration and duration of treatment should be briefly indicated here, I think, even though details are already provided in materials and methods.

The two sentences in results 2.1 could fit in the results 2.2, I think.

Line 158: this title is unclear to me (independent/dependent), and I suggest to change it.

Figures 3, 4 and 5: Images of better quality should be provided, so that the text could become legible.

Each abbreviation should be explained (for example, “SES” line 505).

Table 2: It is not clear which trait is associated with which QTL. To solve this problem, I suggest to add black lines to separate the different traits. Moreover, each abbreviation (PVE, ICIM…) should be explained in the table legend.

Line 271: a reference should be added to support this statement.
